# TritonGym: A Benchmark for Agentic LLM Workflows in Triton GPU Code Generation

## Abstract

Large language models (LLMs) can already draft plausible Triton kernels, yet most existing evaluations still focus on single-shot generation and underplay tool use and feedback. We introduce TritonGym, a benchmark and orchestration framework for evaluating agentic workflows in GPU code generation. TritonGym standardizes access to tools via a function-call API, separating intrinsic model capability from workflow design and enabling fair, apples-to-apples comparison. The benchmark spans a maintained operator set, community samples, out-of-distribution tasks, and DSL extensions, ensuring both generality and extensibility. By providing a common orchestration and evaluation framework, TritonGym democratizes the development of GPU coding agents, supports practical adoption of agent-generated kernels, and facilitates progress on advanced agentic systems.

## 1 Introduction

Efficient GPU kernels are crucial for scaling modern AI (Chetlur et al., 2014). Recent models can already generate hardware-aware kernels that reduce computation time, cost, and energy consumption(Jiang et al., 2024). But even with those gains, many practical challenges remain: ensuring correctness, tailoring kernels to specific hardware, exploiting architectural constraints, and optimizing beyond naive generation. To meet these challenges, agentic workflows offer a better paradigm(Dong et al., 2025). Rather than producing code in one shot, systems using agentic workflows plan their steps, invoke tools (e.g. compilers, verifiers), receive feedback, and iterate through propose-compile-verify-evaluate loops(Huang et al., 2023).

While a developer or LLM planner can launch all sorts of agentic workflows, current GPU code-generation benchmarks(Li et al., 2025; Wang et al., 2025b; Ouyang et al., 2025; Wen et al., 2025) fail to provide a fair and isolated playground to test and benchmark those workflows. The reason is that they treat tool invocation as an internal feature of each evaluated system, so agents differ in what they can compile, run, or verify during generation, confounding comparison with private tooling and ad hoc budgets. *We argue that the ability to invoke code generation tools should be standardized by the benchmark via a function-call API for a fair comparison, not bundled inside each agent.* The benchmark shall provide an interface exposing identical tools, inputs, and budgets to all participants, enabling apples-to-apples evaluation of agentic workflows and cleanly separating intrinsic model capability from workflow design. Without such standardization, progress in agentic code generation remains hard to measure and misaligned with real engineering practice.

However, the introduction of code generation tools raises challenges around benchmark design and evaluation. *The first challenge is to guarantee generality of the benchmark because exposure to search tools may trivialize the benchmark with well-known operators appearing online.* The agent can surface web search and retrieve the implementation of target operators, reducing the task to memorization rather than reasoning and synthesis. At worst, the agent can directly find the reference implementation online, making the benchmark meaningless. *The second challenge is to keep the extensibility of the benchmark when explicit hardware-dependent tools are available.* While practical GPU programming is target-specific and evolves rapidly with hardware advancements, previous benchmarks tend to incorporate fixed platforms with backend knowledge encoded implicitly in the dataset. However, with the reasoning capability of agents and evaluation tools accessing hardware directly, the benchmark must adapt to the unique characteristics of different hardware architectures and programming languages or DSLs (e.g., TLX (Meta, 2025), Gluon (OpenAI, 2020)) to generate

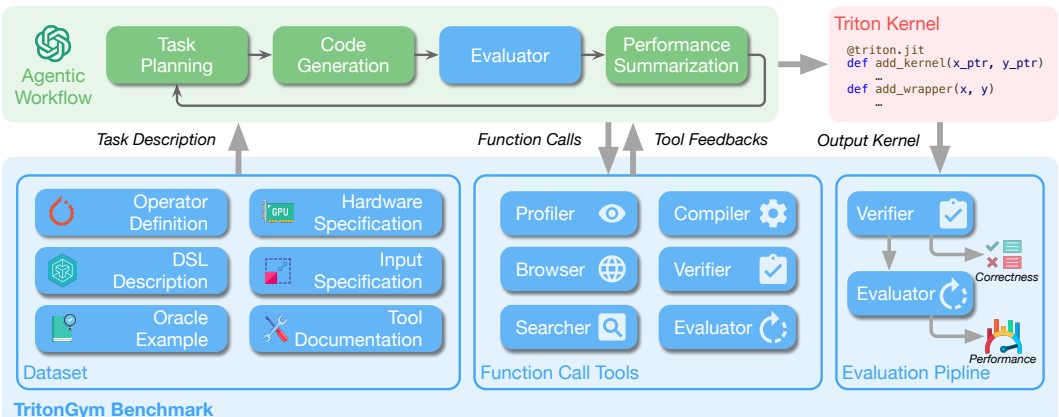

Figure 1: An overview of TritonGym.

efficient, correct code(Tschand et al., 2025). Existing benchmarks often fixate on a single platform or demand universal implementations despite substantial differences across DSLs and backends.

As such, we introduce TritonGym to close this gap with a tool-centric, interactive benchmark featuring generality and extensibility for Triton GPU kernel generation. Its standardized function-call interfaces allow agents to invoke correctness verifiers, performance evaluators, and other tools during generation for feedback to guide their generation. This enables systematic assessment of agentic workflows and paves the way for robust, adaptable code generation systems. To address these generality and extensibility challenges, TritonGym includes novel out-of-distribution (OOD) samples and explicit specifications across backends. We design a specific set of OOD problems with invented, benchmark-only operators whose semantics are specified but for which no public reference implementations exist. As such, success generation requires reasoning and synthesis rather than memorization. To keep the task extensible, TritonGym exposes backend and language descriptors to the agent and supports pluggable backend and DSL extensions, enabling target-appropriate, practical implementations and apples-to-apples comparisons across diverse hardware.

Our empirical evaluation results with TritonGym show that frontier LLMs can synthesize competitive Triton kernels for familiar operators in one-shot generation, yet their accuracy and speed degrade markedly on new backends and out-of-distribution operators that require reasoning beyond memorization. Agentic workflows improve correctness and performance, yet still leave meaningful headroom to carefully engineered baselines. This implies LLM-based GPU coding is far from completed. Besides, with the gym-style playground of TritonGym, we also explore the composition of LLM models, suggesting the advantage of ensembling heterogeneous models in agentic workflows. This evaluation results show that TritonGym provides a stable and forward-looking yardstick for evaluating agentic workflows in GPU kernel generation. This could help the adoption of agent-generated GPU kernels in practice and facilitate the development of advanced agentic systems.

## 2 BACKGROUND

### 2.1 TRITON LANGUAGE AND KERNELS

**Triton** (Tillet et al., 2019) is a Python-like domain-specific language (DSL) for GPU programming particularly well-suited for LLM as a generation target. It balances programmer productivity and high performance by enabling users to write custom GPU kernels more easily than with traditional CUDA or low-level APIs, while still exposing sufficient control for manual optimization. With Triton hiding trivial optimization details of intra-thread communication and memory management, developers can focus on high-level algorithm design and optimizations such as tiling, blocking, and memory hierarchy. This makes it possible to leverage LLM's strong reasoning and planning capabilities while leaving low-level details to the compiler.

To meet the demand of evolving hardware architecture and programming models, a lot of language extensions is built on Triton, making it challenging for coding agents. For example, **Triton Low-Level Extensions (TLX)** (Meta, 2025) adds warp-aware intrinsics, register-backed accumulators,

```
{"add":
  { "torch_operator": "def torch_add(input,
other): …",
    "input_shapes": [
      [[1024, 1024], [1024, 1024]], [[4096,
4096],[4096, 4096]]],
    "triton_operator": "def add_kernel(in_ptr0,
in_ptr1, out_ptr, n_elements, BLOCK_SIZE,): …"
  },
… }
```

(a) Operator description

```
{"tools": [
  { "name": "evaluator",
    "signature": "evaluator(src: str, op: str) -> float",
    "purpose": "Compile and execute the generated Triton function triton_<op>
and return its average latency in milliseconds.",
    "inputs": {
      "src": { "type": "string", "description": "…" },
      "op": { "type": "string", "description": "…" }},
    "returns": { "type": "float", "units": "ms", "description": "…" }
  }, …]}
```

(b) Tool documentation

```
{"tlx":
  " TLX (Triton Low-level Language Extensions) is a low-level, warp-aware, hardware-near extension of the Triton DSL. It offers
  intrinsics and warp-specialized operations for fine-grained GPU control, hardware-oriented primitives for advanced kernel
  development, and explicit constructs for GPU memory, computation, and asynchronous control flow. TLX is designed for expert users
  pushing Triton closer to the metal.
  - `tlx.async_tasks` and `tlx.async_task`
  ```python
    with tlx.asycn_task(num_warps = 4)
      ...
  ```
  `tlx.async_task(num_warps=4)` defines a warp-specialized asynchronous task that explicitly reserves 4 warps in addition to those
  used by the trunk task. ",
… }
```

(c) Language description

```
{"GPUs": [
  { "name": "H100_SXM5",
    "process": "4nm",
    "transistor_count": "80 billion",
    "memory": {
      "type": "HBM3",
      "capacity": "80 GB",
      "bandwidth": "3000 GB/s"
    },
    "vector_performance": {
      "fp64": "30 TFLOPS", …},
    "tensor_performance": {
      "fp64": "60 TFLOPS", …},
    "SM_num": "132",
    "GPC_num": "7",
    "on_chip_mem": {
      "LLC_size": "50MB",…},
    "compute_capability": {
      "thread_per_warp": 32, …},
    "power_consumption": "700W"
  },
…]
}
```

(d) Hardware specification

Figure 2: Dataset design of TritonGym.

pipeline execution, and finer synchronization primitives to better exploit modern GPU features. **Gluon** (OpenAI, 2020) is a lower-level dialect within the Triton compiler stack, exposing primitives for layout encoding and more explicit control over encoding and lowering to better target hardware primitives. The code generation system must be extensible to understand these emerging extensions to generate efficient, correct code tailored to specific hardware and optimization goals.

## 2.2 AGENTS FOR CODE GENERATION

Agents with LLMs has been a promising code generation solution(Dong et al., 2025), where they not only generate code but also interact with tools, execute code, and iteratively improve their outputs based on feedback. This agentic paradigm leverages the LLM's ability to reason over multiple steps, use external resources (e.g., compilers, profilers), and adapt its behavior through reflection and optimization loops. More broadly, agentic evaluation is becoming common in code generation research, as it better reflects real-world workflows where iterative improvement and tool use are essential for producing high-quality, correct, and performant code. For example, **AlphaEvolve** (Novikov et al., 2025) introduces an evolutionary agent framework for code generation, combining search, self-play, and performance feedback to progressively refine generated programs. Similarly, AutoEvolveNP (Yu et al., 2025) explores how agentic LLMs can autonomously evolve code solutions for computationally hard problems, demonstrating the potential of iterative self-improvement loops.

In the context of GPU kernels, the agentic system design becomes salient due to the complexity of hardware architectures, the need for performance tuning, and the iterative nature of optimization. **Geak** (Wang et al., 2025a) introduces an agent pipeline consisting of a generator that proposes code, an evaluator that tests correctness and performance, a reflector that analyzes failures, and an optimizer that refines the solution. Building on this direction, **Astra** (Wei et al., 2025) advances the workflow by coordinating specialized agents for analysis, scheduling, and performance tuning, demonstrating substantial speedups via division of labor and learned heuristics. These works underscore the growing importance of agentic paradigms in code generation and motivate benchmarks that can evaluate such capabilities in specialized domains like GPU kernel synthesis.

## 3 TRITONGYM BENCHMARK

TritonGym is composed of three key components as shown in Fig. 1: (1) a diverse and extensible dataset of GPU kernel synthesis tasks, (2) a suite of function-call tools that enable agentic workflows, and (3) a rigorous evaluation routine that measures correctness and performance under realistic constraints. In this section, we describe each component in detail.

### 3.1 DATASET DESIGN

#### 3.1.1 DATASET FORMAT

The Triton code generation task in TritonGym is formulated as synthesizing a Triton kernel that implements a specified operator given its precise semantics, input shapes, and a reference PyTorch

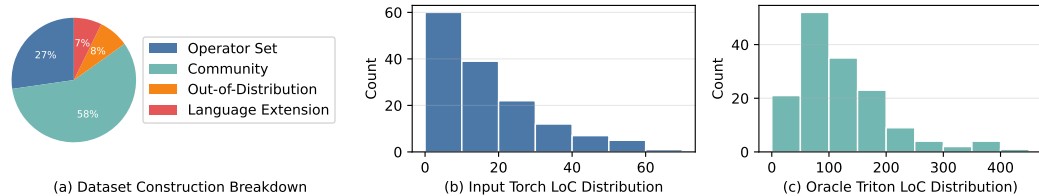

Figure 3: Dataset construction of TritonGym. Left: source of samples. Right: LoC distributions.

implementation. For performance optimization, we also provide extensible hardware and DSL descriptions to guide the agent toward target-appropriate implementations. These components are structured as JSON files as shown in Fig. 2.

**Task Formulation.** TritonGym formulates the code generation task with operator definition (`torch_code`) and input specification (`input_shapes` and `data_types`). An oracle example (`triton_operator`) establishes reference performance developed by human experts and indicates headroom for optimization. The bundle also carries metadata such as domain, backend, and language to retrieve the relevant descriptions used for this task.

**Backend Extensions.** The benchmark is extensible across backends and DSLs via explicit descriptors available to the agent at generation time. For hardware specification, we provide a JSON file that the agent can condition on when choosing tiling, vector widths, memory layouts, and synchronization. The JSON records the GPU model and micro-architecture, memory hierarchy and bandwidth, etc as shown in Fig. 2-(d). Because the format is extensible, additional accelerators or backend-specific keys can be added without breaking existing workflows, while agents remain free to exploit the information to produce practical, target-aware kernels.

**Language Description.** Complementing the backend specification, each task may include a compact DSL descriptor that makes the language surface area explicit (Fig. 2-(c)) and a description that records the DSL usage. This includes the intended programming model (e.g., block-level SPMD with predication), common scheduling and tiling idioms (grid mapping, vector widths), and available intrinsics and memory primitives (e.g., `load`/`store`, `barriers`). Exposing these capabilities lets agents reason about which transformations are legal and performance-relevant on the target, while remaining agnostic to implementation details. The same operator task can thus be evaluated across different DSLs by swapping descriptors without changing the underlying semantics.

### 3.1.2 DATA COLLECTION

We construct the TritonGym benchmark set by combining the samples from several sources as shown in Fig. 3-(a). Specifically, we maintain a high-quality operator set for common practices, collect a wide spectrum of operators to cover novel applications, and use the OOD and DSL extension samples to push the frontier of generalization.

**Maintained Operators Set.** We maintain a curated set of high-quality Triton implementations for commonly used operators covering core ML applications. The operator set aggregates SOTA Triton kernels and vetted external implementations in other sources (such as CUTLASS(NVIDIA Corporation, 2025), ThunderKitten(Spector et al., 2024)) as reference, and we will open-source it to facilitate future research. In TritonGym, these maintained operators are a primary source of samples with canonical PyTorch references, and normalize names and interfaces to TritonGym's schema.

**Community Samples.** We also curate operators from public repositories and prior code-generation benchmarks. Each sample is normalized into our task format with precise operator semantics, input-shape specifications, and a PyTorch reference implementation that serves as the correctness oracle. This track captures practical kernels that agents are likely to encounter and enables direct comparison to previously reported results within our unified evaluation harness.

**Out-of-Distribution Operators.** To evaluate whether agents truly internalize optimization principles rather than memorizing reference kernels, we design OOD operators with novel, benchmark-only semantics for which no public implementations exist. These operators are constructed by altering the underlying operator, for example, replacing summation with multiplication. Such folds are mathematically natural and potentially useful in practice, even though they are absent from standard libraries. By focusing on operators that are structurally similar yet unseen, we can test whether

LLMs learn the essence of GPU optimization, such as scheduling, tiling, and memory management, independently of the specific operation being reduced. For these tasks, we specify the PyTorch oracle from first principles and handcraft efficient Triton implementations as baselines. This pairing prevents leakage through web search while providing a meaningful performance target for agents to surpass. The demonstration of these operators is shown in § C.

**DSL Extension Samples.** To exercise portability beyond a single DSL, we also include samples that target language extensions for TLX and Gluon. These tasks are selected to stress the features exposed in the corresponding language descriptors (Fig. 2-(c)) and are more challenging to program compared to the base Triton DSL due to their additional abstraction and optimization capabilities, such as asynchronous behaviors and explicit barriers. By pairing identical operator specifications with different DSL descriptors, TritonGym isolates the effect of DSL capabilities from task semantics and demonstrates that agentic workflows transfer across languages.

Fig. 3- (b) and (c) plot line-of-code(LoC) distributions for the PyTorch reference and the Triton oracle, respectively. While relatively short and easy samples (PyTorch LoC $\leq 20$ and Triton LoC $\leq 100$) take a large portion, TritonGym includes challenging tasks with over 50 LoC PyTorch definition and require over 300 LoC in Triton to implement by human experts.

### 3.2 Tools with Function Call in Agentic Workflow

We introduce the standard tools supported in the TritonGym framework, facilitating agentic code generation, and show how they can express workflows from prior work. These tools convert kernel synthesis from a one-shot act into a closed-loop process with environmental feedback.

#### 3.2.1 Introduction of Tools

At the core of our tool suite are three execution-time services that enforce correctness before performance: a compiler, a verifier, and an evaluator. The `compiler(src: str, op: str) -> str` checks that the provided Python source defines a callable function and that it compiles without syntax or symbol errors. It returns an empty string on success and a diagnostic message otherwise. This immediate structural feedback is crucial for rapid iteration, allowing agents to resolve import issues, naming mismatches, and syntax mistakes before investing effort in optimization.

Once the code is compiled, the probabilistic `verifier(src: str, op: str) -> bool` compares outputs of the operator against the PyTorch reference on shapes specified by the task with random materialized tensors for N times. Verification succeeds when both absolute and relevant difference is controlled in pre-defined thresholds. By enforcing semantic correctness early, the verifier prevents misleading speedups from incorrect kernels.

For performance assessment, the `evaluator(src: str, op: str) -> float` compiles and executes the generated operator and returns average latency, reporting `NaN` on compilation or runtime failure. This quantitative signal closes the optimization loop where agents can explore tiling, memory layouts, and parallelism strategies and use measured latency to drive search or decision policies.

We also provide retrieval tools for the agents to acquire background knowledge or performance insights to guide their optimization. The `searcher(key: str) -> list[dict]` issues a web search and returns parsed results, and the `browser(url: str) -> dict` fetches a page and extracts text, links, and metadata. These tools help agents acquire background knowledge—API nuances, DSL idioms, or hardware-specific tips—that accelerate design. Importantly, our benchmark mitigates leakage by including out-of-distribution operators with novel semantics and by evaluating on shapes and settings where naive copy-paste is insufficient; retrieval therefore informs reasoning without trivializing the task. `profiler(src: str, op: str) -> str` launches the NCU profiler and returns a structured report of kernel execution metrics, enabling agents to identify bottlenecks such as memory stalls, low occupancy, or divergent warps and to target their optimizations accordingly.

Together, these tools enable disciplined feedback loops, optionally augmented by targeted retrieval, which mirrors how human experts develop high-performance GPU kernels and makes agentic code generation both measurable and reproducible in TritonGym.

#### 3.2.2 Expressiveness of Workflows

Our function-call interface is expressive enough to implement the agentic workflows proposed in prior works, from single-shot generation to multi-step planning with retrieval, compilation, verifica-

Figure 4: Agentic code generation workflows from previous works within TritonGym.

tion, and performance-driven refinement (Fig. 4). For example, the AlphaEvolve workflow can be implemented by launching the ensembling LLM and evaluator iteratively. The GeakAgent and Astra workflows include the verifier but consume the verification results to guide the next step differently.

Besides, implementing diverse workflows within the same harness guarantees a fair comparison. All methods operate over the identical task set, shape seeds, and oracle definitions and they share the same compiler, verifier thresholds, and evaluator configuration. In the benchmark, we record every tool invocation and outcome, making the propose-compile-verify-evaluate loop reproducible and comparable across methods. As a result, differences in accuracy or latency reflect the workflow itself rather than incidental variations in datasets, test harnesses, or measurement procedures.

## 3.3 EVALUATION ROUTINE

### 3.3.1 EVALUATION METRICS

We use two commonly used metrics to measure the correctness and performance of the synthesized operators by the agentic workflows in the evaluation routine.

**Correctness.** The correctness metric **Pass@K** counts how many trials succeed in a total K times as, $\text{Pass@K} = \sum_{t=1}^{K} v_t$. This probabilistic verification is considered the best practice for numerical kernels due to the deterministic and shallow control flow (**?**). For a fixed operator instance, let $v_t \in \{0, 1\}$ indicate whether trial $t \in \{1, \dots, K\}$ passes the probabilistic verifier, for each required shape, $N$ random input draws are generated (with fixed seeds) and all must satisfy the error thresholds.

**Performance.** We use **Perf@K** to measure best performance relative to the oracle over only the verified-correct trials in K times as, $\text{Perf@K} = \max_{t : v_t=1} \ell^{\text{oracle}}/\ell_t$. Let $\ell_t$ denote the evaluator-reported average latency (in milliseconds) for trial $t$, and let $\ell^{\text{oracle}}$ be the oracle latency from the provided reference implementation. If no trial is correct (i.e., $\sum_t v_t = 0$), we set $\text{Perf@K} = 0$ for that instance. As a result, values above 1 indicate better implementation over the human oracle Triton code and values below 1 are slower. For performance measurement, we use the average latency for repeated runs as the default metric while providing an interface for users to plug in customized metrics such as memory footprint or TFLOPs calculation function for particular operators.

### 3.3.2 LEADERBOARD

We open-source the dataset and evaluation suite with a Docker image to encourage community contributions. We also host a website that serves as the submission portal and leaderboard hub for TritonGym. All submissions are executed in the same evaluation harness described above and are reported with the same metrics and configuration, enabling transparent comparison across models and workflows. We maintain two complementary leaderboards. The one-shot leaderboard isolates intrinsic LLM capability by running each model in a single pass. The agentic leaderboard evaluates complete workflows under a fixed trial budget and tool policy. In both cases, entries are ranked primarily by dataset-averaged correctness and secondarily by performance.

## 4 TRITONGYM BASELINE EVALUATION

To demonstrate the utility of TritonGym, we investigate a suite of state-of-the-art LLMs and agentic workflows. Our evaluation focuses on three key aspects: (1) the ability of models to generate correct and efficient Triton kernels, (2) the effectiveness of agentic workflows in improving code quality through iterative refinement, and (3) the generalization capabilities of models to handle OOD tasks and adapt to different backends and languages.

Table 1: Evaluation results of baseline workflows and LLM models.

| | Standard Samples | | OOD Samples | | DSL Samples | | Full Benchmark | |
|---|---|---|---|---|---|---|---|---|
| | Pass@1 | Perf@1 | Pass@1 | Perf@1 | Pass@1 | Perf@1 | Pass@1 | Perf@1 |
| **One-shot** (avg.) | 35.5% | 0.322 | 15.4% | 0.143 | 14.6% | 0.062 | 32.8% | 0.292 |
| + chatgpt-4o | 20.7% | 0.164 | 7.7% | 0.023 | 8.3% | 0.025 | 18.7% | 0.141 |
| + llama-4-maverick | 26.8% | 0.205 | 7.7% | 0.063 | 16.7% | 0.053 | 24.6% | 0.182 |
| + deepseek-chat | 39.4% | 0.350 | 23.1% | 0.269 | 16.7% | 0.081 | 36.1% | 0.327 |
| + claude-sonnet-3.5 | 54.9% | 0.569 | 23.1% | 0.217 | 16.7% | 0.087 | 50.0% | 0.512 |
| **Geak** (avg.) | 39.2% | 0.348 | 21.2% | 0.250 | 25.0% | 0.113 | 36.7% | 0.323 |
| + chatgpt-4o | 25.0% | 0.221 | 23.1% | 0.111 | 33.3% | 0.148 | 25.4% | 0.210 |
| + llama-4-maverick | 33.3% | 0.227 | 7.7% | 0.066 | 16.7% | 0.056 | 29.7% | 0.202 |
| + deepseek-chat | 40.9% | 0.363 | 30.8% | 0.548 | 25.0% | 0.130 | 38.1% | 0.362 |
| + claude-sonnet-3.5 | 57.5% | 0.582 | 23.1% | 0.273 | 25.0% | 0.118 | 53.9% | 0.528 |
| **AlphaEvolve** (avg.) | 45.1% | 0.368 | 28.9% | 0.353 | 25.0% | 0.088 | 42.3% | 0.346 |
| + chatgpt-4o | 30.4% | 0.196 | 15.4% | 0.209 | 16.7% | 0.040 | 28.4% | 0.189 |
| + llama-4-maverick | 38.7% | 0.217 | 15.4% | 0.069 | 16.7% | 0.093 | 35.1% | 0.197 |
| + deepseek-chat | 43.3% | 0.373 | 23.1% | 0.348 | 33.3% | 0.095 | 41.9% | 0.351 |
| + claude-sonnet-3.5 | 67.9% | 0.687 | 61.5% | 0.785 | 33.3% | 0.125 | 65.1% | 0.651 |
| **AlphaEvolve + Interchange** | 58.6% | 0.389 | 38.5% | 0.202 | 25.0% | 0.038 | 54.5% | 0.349 |

## 4.1 EXPERIMENTAL SETUP

**Models.** We evaluate SOTA LLMs from multiple providers as backbones using their respective APIs with consistent prompt setting, temperature, and decoding settings to ensure fair comparison. Specifically, we adopt LLM APIs with the following versions: claude-3.5-sonnet-20240620, gpt-4o-2024-08-06 (Achiam et al., 2023), Llama-4-Maverick-17B-128E-Instruct-FP8 (Meta, 2024) and deepseek-chat-v3.0 (Liu et al., 2024).

**Workflows.** We reimplement representative agentic strategies within TritonGym's framework including AlphaEolve (Novikov et al., 2025) and Geak (Wang et al., 2025a) mimicking their LLM-tool interaction workflows. Using the function-call APIs in TritonGym, one can easily implement other agentic workflows and prompt settings. We give a detailed explanation of the agent settings and prompt designs in § A. Using the function-call APIs in TritonGym, one can easily extend these implementations to other agentic workflows and prompting strategies, while keeping the evaluation pipeline and logging unified.

## 4.2 ONE-SHOT GENERATION RESULTS

In the one-shot track, models often produce reasonable kernels for familiar operators, but they plateau below expert performance. As summarized in Tbl. 1, standard samples (including community and maintained sets) are substantially easier than OOD and DSL categories, where success is highest on familiar operators and drops when the operator deviates from seen templates, the backend changes, or a language extension is required. The trend is consistent across models, when one-shot succeeds where prior exposure suffices, but without tool feedback, it struggles to repair compile/runtime issues and to pass verification on edge cases. Overall, one-shot captures baseline competence yet leaves headroom that motivates agentic workflows.

*Observation 1:* One-shot generation is competitive on familiar operators but degrades on OOD and DSL tasks, underscoring limits of memorization and the need for agentic workflows.

## 4.3 AGENTIC WORKFLOW RESULTS

For the agentic workflows shown in Tbl. 1, the results show a consistent pattern: agentic workflows outperform one-shot across base models, with the largest gains under OOD or DSL settings. Iterative feedback turns many non-building programs into runnable ones and then uses feedback to refine scheduling, tiling, and boundary handling, lifting both correctness and performance. Stronger base models push standard tasks close to saturation with agents, while lighter bases still benefit but remain capacity-limited. We also observe complementary behaviors, for instance, Geak tends to eliminate compile failures quickly and secure early wins, whereas AlphaEvolve continues improving with additional trials and recovers deeper performance. Across models and categories, the trend is monotonic improvements over one-shot and better robustness to new operators and backends.

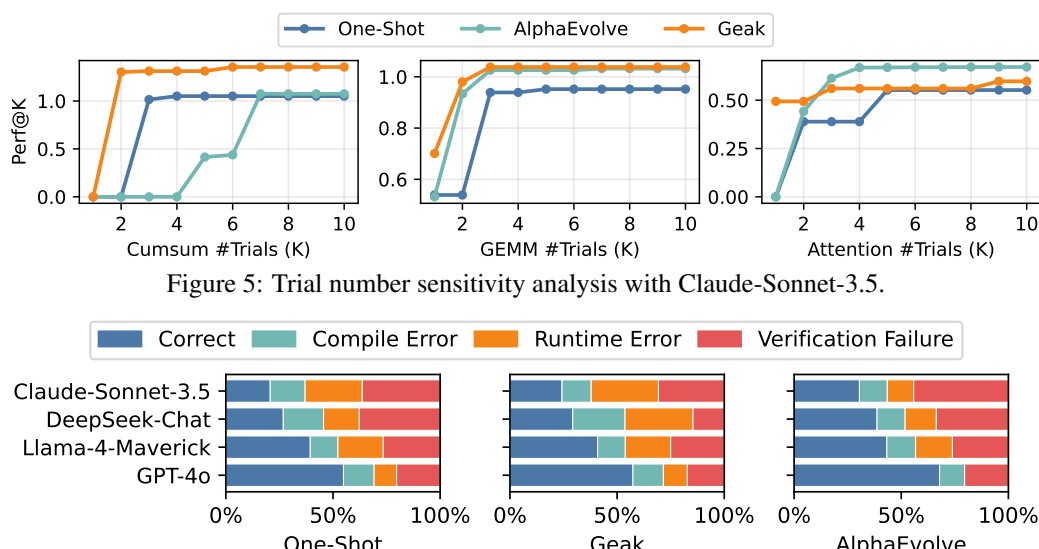

Figure 5: Trial number sensitivity analysis with Claude-Sonnet-3.5.

Figure 6: Generation failures breakdown.

*Observation 2:* Agentic workflows substantially improve correctness and performance, illustrating the benefit of feedback-driven iteration.

## 4.4 TRIAL BUDGET ANALYSIS

Fig. 5 shows how performance scales with the trial budget $K$ across three representative operators. Specifically, we align the results of one-shot with $K$ trials and agents with $K$ iterations within one trial. For a trivial operator such as cumsum, agentic workflows quickly dominate: a few iterations suffice to turn early failures into verified kernels, and they remain ahead of one-shot at every $K$. For a well-studied operator like GEMM, agents rapidly converge to expert-level performance, whereas one-shot plateaus below that ceiling even with more trials; the propose–compile–verify–evaluate loop reliably rediscovers the tiling and scheduling patterns used by human implementations. For more difficult cases (e.g., attention), agents continue to improve with additional trials yet still leave headroom, but they consistently outperform one-shot across the full budget range. Overall, agentic search translates extra trials into steady gains, while one-shot saturates early at a lower ceiling.

*Observation 3:* With the same trial budget, agentic workflows outperform one-shot by a large margin, and the design of such LLM-tool interaction is crucial to the final performance.

## 4.5 CASE STUDIES

**LLM Interchange Orchestration.** To showcase the gym-style workflow composition of TritonGym, we test a hybrid AlphaEvolve agent that orchestrate Llama-4 and DeepSeek-chat model interchangeably between different steps denoted as *AlphaEvolve+Interchange* in Tbl. 1. The result shows that the mixed invocation of models from different vendors as the same agent role achieves a higher Pass@1 compared to using a homogeneous model alone. We hypothesize that this is because the divergence of different models leads to a larger search space. For example, each model tends to stick to similar mistakes, yet two models can produce solutions that differ from each other as a mutual reference. However, this interchange may interfere with each model's reasoning of performance improvement, leading to unstable Perf@1. Due to the topic of this work, we do a preliminary experiment on this phenomenon and leave further exploration to future work. This kind of exploration highlights the effectiveness of the proposed gym-style benchmark.

**Failure Breakdown.** Fig. 6 decomposes outcomes into correct, compile, runtime, and verification failures. One-shot exhibits substantial compile and verification failures, reflecting frequent API/syntax issues and mismatches to the oracle. Geak markedly reduces compile errors with its iterative compile–fix loop, turns many non-building programs into runnable ones, yet correspondingly shifts mass into runtime or verification as more candidates execute but still miss full specification. AlphaE-

Table 2: Comparison of code generation benchmarks.

| Benchmark | Input | Output | Out-of-Distribution Examples | Backend Extension | Language Extension | Function Call |
|---|---|---|---|---|---|---|
| TritonBench(Li et al., 2025) | Natural Language | Triton | ✗ | ✗ | ✗ | ✗ |
| Geak(Wang et al., 2025b) | Natural Language | Triton | ✔ | ✗ | ✗ | ✗ |
| KernelBench(Ouyang et al., 2025) | PyTorch | Any | ✗ | ✗ | ✗ | ✔ |
| MultiKernelBench(Wen et al., 2025) | PyTorch | Backend Specific | ✗ | ✔ | ✗ | ✗ |
| **TritonGym (this work)** | PyTorch | Triton | ✔ | ✔ | ✔ | ✔ |

volve goes further by using feedback to refine kernel structure and scheduling, it reduces runtime failures and increases the share of correct solutions. Across the evaluated models, agentic workflows principally convert compile-time failures into executable programs, then improve execution correctness. And verification remains the bottleneck, motivating richer oracles/specs and stronger feedback signals for correctness to unlock further gains.

**Generated Kernel with DSL Extension.** Fig. 7 shows a case study of a kernel generated by the AlphaEvolve agent of a pipelined GEMM operator using TLX. This generated kernel uses Hopper architecture-specific features, including asynchronous TMA transfers from global to shared memory, multi-stage buffering with commit/wait groups, and `mma` instructions of the Tensor Core unit, which are essential to approach Hopper throughput. One-shot LLMs consistently fail to produce a correct kernel in a single decode pass. In contrast, agentic workflows succeed after several trials. This stepwise construction with feedback is essential to

```
# Main computation loop
for k in range(0, tl.cdiv(K, BLOCK_K)):
    ...
    # Wait for current buffer to be ready
    tlx.async_load_wait_group(NUM_STAGES - 2)
    # Perform async dot product
    acc = tlx.async_dot(buffer_a, buffer_b, acc)
    acc = tlx.async_dot_wait(NUM_STAGES - 1, acc)
    ...
    # Prefetch next buffer
    token_a = tlx.async_load(a_ptrs, next_buffer_a)
    token_b = tlx.async_load(b_ptrs, next_buffer_b)
    tlx.async_load_commit_group([token_a, token_b])
    ...
```

Figure 7: Simplified pipelined GEMM generated by AlphaEvolve with TLX.

approach high performance for these emerging platforms and it only emerges when the workflow can read compiler/verifier feedback and iteratively revise the kernel.

## 5 RELATED WORK

Several benchmarks have been proposed to evaluate GPU code generation capabilities. **TritonBench** (Li et al., 2025) targets Triton kernels with synthesized natural language as input, focusing on real-world workloads such as GitHub and PyTorch operator implementations. It evaluates both call accuracy (matching API signatures) and execution accuracy (runtime correctness), as well as GPU efficiency. However, TritonBench does not incorporate agentic workflows or support iterative feedback loops. **Geak** (Wang et al., 2025b) ports TritonBench to AMD backends and introduces a novel agent pipeline comprising generator, evaluator, reflector, and optimizer modules. This design explicitly supports agentic workflows, allowing for iterative code generation, evaluation, and optimization. However, the executed tools are handled by the agent developer with ad-hoc handling, limiting their reproducibility and preventing them from being benchmarked systematically with other agentic workflows. **KernelBench** (Ouyang et al., 2025) focuses on GPU kernel generation, allowing the LLM to dump any language including Triton. While it provides a rigorous evaluation of static code outputs, its design is limited in supporting agentic workflows, there is little emphasis on iterative code refinement or tool use beyond initial code generation. **MultiKernelBench** (Wen et al., 2025) expands the scope to multiple platforms, including CUDA, AscendC, and Pallas as target languages for each platform, enabling evaluation across different hardware and APIs. Its strength lies in the breadth of coverage, but the benchmark primarily uses static evaluation and does not accommodate agent-based or iterative code improvement.

## 6 CONCLUSION

TritonGym proposes a reproducible benchmark for automatic Triton kernel synthesis that targets agentic workflows across a wide spectrum of operators. With carefully designed task formulation, the benchmark is extensible to new backends and DSLs, and builds a fair arena for evaluating the power of agentic workflows instead of just the capability of LLMs. This opens a new direction in assessing how LLMs reason, plan, and adapt in high-performance code generation.

REPRODUCIBILITY STATEMENT

We release all artifacts needed to reproduce our results. We have two open-source repositories: one for the TritonGym framework and one for the maintained Triton operator set and benchmark suite. We also have a public leaderboard that evaluates submissions with the same harness to support independent verification. To preserve anonymity during review, we omit links here and the URLs for the evaluation framework, operator set, and leaderboard will be included in the camera-ready version.

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

## A  AGENT SETTINGS

**One-shot** is the simplest setting: the agent constructs a single prompt that embeds the PyTorch operator specification and optional instructions, queries the LLM once, and decodes the generated text into executable Triton code. No iterative refinement or evaluation feedback is performed, making it suitable for straightforward operators but less robust to initial generation errors.

**AlphaEvolve** extends this into an evolutionary workflow. At each attempt, the agent samples inspirations (successful or failed trials) from a prompt database and augments the base prompt with code diffs and evaluation results. The LLM then proposes a new candidate, which is automatically compiled, verified, and benchmarked by the evaluator. Feedback such as compilation errors or performance metrics, is stored back into the database, guiding subsequent attempts. After a fixed number of iterations (default: 5 attempts with 3 inspirations each), the agent selects the best candidate, prioritizing correctness and then latency. This workflow captures the essence of evolutionary search with evaluation-in-the-loop refinement.

**Geak** implements a more structured multi-agent workflow with both sequential and parallel scaling. Within a sequential run, the process alternates between (i) a *Generator* that produces Triton code conditioned on hardware-specific optimization rules, (ii) an *Evaluator* that conducts cascaded functionality and performance tests, (iii) a *Reflector* that analyzes errors and suggests fixes, and (iv) an *Optimizer* that proposes performance improvements when functionality passes. The sequential loop runs for at most 3 iterations, with up to 3 debugging attempts allowed before resetting. In parallel mode, multiple such sequential runs (default: 1–4) are executed concurrently, and the best-performing code among successful runs is selected.

In the following, we demonstrate the prompts used in our baseline evaluation to reimplement the one-shot generator, Geak, and AlphaEvolve agents. Each prompt is designed to guide the LLM towards generating high-quality Triton kernels while adhering to specific rules and leveraging relevant knowledge.

## One-shot Generator Prompt

You are an expert in writing high-performance GPU kernels using Triton.
Your task is to iteratively improve the provided PyTorch operator by generating a Triton implementation.
Rules:
- Focus on correctness first; then optimize for speed (tiling, blocking, memory hierarchy).
- Return ONLY executable Python code (no markdown, no commentary).
- Name the function `triton_<operator_name>` where `<operator_name>` matches the reference `torch_<operator_name>`.
- The function signature must match the reference exactly.
- If you modify code, use a SEARCH/REPLACE diff style when possible.
- If the previous version failed, explicitly fix the cause while keeping other parts intact.
# PyTorch function specification:
`{code}`

## Geak Knowledge Prompt

# Triton Optimization Knowledge:
- Utilize shared memory for frequent data access within a thread block.
- Ensure memory coalescing: consecutive threads should access consecutive memory addresses.
- Use vectorized loads/stores (`tl.load`/`tl.store`) where possible.
- Be mindful of bank conflicts in shared memory.
- Use tiling to exploit data locality.
- Prefer explicit masking (e.g., `mask = idx < size`) for boundary checks.
- Kernel signature must include `@triton.jit` decorator and use `tl.constexpr` for compile-time constants.
`{more specific knowledge}`

## Geak Generator Prompt

You are an expert in writing high-performance GPU kernels using Triton.
Your task is to iteratively improve the provided PyTorch operator by generating a Triton implementation.
Rules:
- Focus on correctness first; then optimize for speed (tiling, blocking, memory hierarchy).
- Return ONLY executable Python code (no markdown, no commentary).
- Name the function `triton_<operator_name>` where `<operator_name>` matches the reference `torch_<operator_name>`.
- The function signature must match the reference exactly.
- If you modify code, use a SEARCH/REPLACE diff style when possible.
- If the previous version failed, explicitly fix the cause while keeping other parts intact.
# PyTorch function specification:
`{code}`

## Geak Reflector Prompt

You are a debugging expert. Analyze the Triton kernel code and the error message. Propose concise, specific fixes to resolve the issue. Focus on the root cause.
# Triton Kernel Code:
`{code}`
# Error Feedback (Compilation/Runtime):
`{error_feedback}`
# Evaluation Result Context:
`{eval_result}`
Analyze the error and suggest the key changes needed to fix the code. Be specific about what to change and why:

---

**Geak Optimizer Prompt**

You are a performance optimization expert for GPU Triton kernels. Suggest specific, actionable optimizations to reduce kernel latency.
# Current Triton Kernel Code (Latency: {latency_ms} ms):
{code}
# Recent Generation History:
{history_context}
Suggest specific optimizations (e.g., better tiling, memory access patterns, using shared memory, vectorization). Focus on the most promising changes:

---

**AlphaEvolve Generator Prompt**

You are an expert in writing high-performance GPU kernels using Triton.
Your task is to iteratively improve the provided PyTorch operator by generating a Triton implementation.
Rules:
- Focus on correctness first; then optimize for speed (tiling, blocking, memory hierarchy).
- Return ONLY executable Python code (no markdown, no commentary).
- Name the function `triton_<operator_name>` where <operator_name> matches the reference `torch_<operator_name>`.
- The function signature must match the reference exactly.
- If you modify code, use a SEARCH/REPLACE diff style when possible.
- If the previous version failed, explicitly fix the cause while keeping other parts intact.
# PyTorch function specification:
{code}

---

**AlphaEvolve Sampler Prompt**

# Inspirations from past trials:
# Past trial {i}:
{code}
# Result:
{result}
Analyze the above past trial carefully. Identify what worked correctly and what caused errors or inefficiencies.
Retain the correct and efficient logic whenever possible, but modify or replace the faulty parts.
Focus on improving kernel latency, scalability across GPUs, and avoiding unnecessary recomputation.
Learn from the past trials: fix correctness errors, preserve working logic, and aim to reduce latency if possible.
Generate a new improved version below:

## B    FUTURE WORK

**PyTorch Integration.** To fasicilitate the practical use of generated kernels, we will provide an automatic integration pipeline that turns a set of LLM-generated Triton kernels into a drop-in PyTorch backend. The pipeline will (i) scaffold a clean folder layout and manifests, (ii) generate binding glue and wrappers, and (iii) run a test battery that reuses PyTorch's operator tests alongside numerics checks and tolerance audits on real shapes drawn from popular HF models. For performance, we will include warmup-aware microbenchmarks and shape suites representative of end-to-end workloads.

**Broader Backend Coverage.** We will extend TritonGym beyond NVIDIA A100/H100 by adding additional backends, for example MI300 from AMD. This effort involves publishing hardware descriptors that capture architectural limits and guidance, and providing backend-related oracle Triton implementation samples. This will help ensure that the benchmark remains relevant as hardware evolves and that models can generalize across different GPU architectures.

**Auto OOD Generation.** We will auto-generate OOD operators by applying random algebraic transformation rewrites with pre-defined operators as templates. This alegebraic transformation can be

defined as several primitives such as commutative, associative, distributive, and normalization transformations (Wu et al., 2025). Then we can random sample a sequence of these transformations to apply to a base operator. With these transformations at high-level PyTorch operator and corresponding Triton code together, we can derive new OOD operators automatically. This approach allows us to systematically create challenging OOD examples that test the model's ability to generalize beyond familiar patterns, without manual effort.

**Ecosystem of Agentic Workflows.** We will include more baselines in the benchmark leaderboard (e.g., Astra (Wei et al., 2025)) to sustain a live benchmark. We will also investigate better fairness controls such as token-budget accounting, memory or retrieval consumption beyond current trial budget metrics.

**Multi-GPU and Distributed Kernels.** We also plan to extend our DSL support to multi-GPU settings, for example with Triton-Distributed DSL extension, and introduce a backend resource manager that can allocate devices, enforce topology-aware placement (NVLink/PCIe), and orchestrate communication backends. Verification will include cross-device determinism checks and scale-out regressions, while performance reporting will add strong/weak scaling and compute–communication overlap metrics. This capability enables evaluating operator classes that are inherently distributed (e.g., sharded attention, mixture-of-experts routing) and brings the benchmark closer to production training and serving scenarios.

## C  OUT-OF-DISTRIBUTION EXAMPLES

We showcase several representative crafted OOD examples in TritonGym that test the model's ability to generalize beyond familiar patterns. These examples include trivial adaptation of normalization techniques and complex graph-level transformations that require deeper reasoning and adaptation.

Table 3: List of OOD examples in the benchmark.

| Op Name | Description | Python Code |
|---------|-------------|-------------|
| Affine GEMM | Performs a GEMM of matrices A and B, where B is generated via an affine transformation: $B_{ij} = \alpha i + \beta j$. | ```k_indices = torch.arange(K,\n    device=a.device,\n    dtype=torch.float32)\nn_indices = torch.arange(N,\n    device=a.device,\n    dtype=torch.float32)\nb_prime = (k_indices[:, None] * alpha\n    + n_indices[None, :] * beta)\nb = b_prime.to(a.dtype)\nresult = torch.matmul(a, b)``` |
| Max Reduce GEMM | Replaces the standard sum-reduction in matrix multiplication with a max-reduction operation. | ```a_expanded = a.unsqueeze(1)\nb_expanded = b.T.unsqueeze(0)\nall_products = a_expanded * b_expanded\nresult = torch.max(all_products,\n    dim=2).values``` |
| GELU Attention | An attention-like mechanism where the Softmax function is replaced by an element-wise GELU activation function. | ```p = torch.matmul(q, k.transpose(2, 3))\np = gelu(p)\np = p.to(ref_dtype)\nref_out = torch.matmul(p, v).half()``` |
| RELU Attention | An attention-like mechanism where the Softmax function is replaced by an element-wise ReLU activation function. | ```p = torch.matmul(q, k.transpose(2, 3))\np = relu(p)\np = p.to(ref_dtype)\nref_out = torch.matmul(p, v).half()``` |

**Table 3 – Continued from previous page**

| Op Name | Description | Python Code |
|---------|-------------|-------------|
| Contiguous GEMM | A simplified attention mechanism where the Softmax activation is removed, resulting in two consecutive matrix multiplications. | ```p = torch.matmul(q, k.transpose(2, 3))```
```p = p.to(ref_dtype)```
```ref_out = torch.matmul(p, v).half()``` |
| Chaos Norm | A custom normalization layer involving a complex sequence of operations, including dynamic epsilon calculation and conditional transformations. | ```n_cols = x.shape[1]```
```max_val = torch.max(x, dim=1,```
```↪ keepdim=True)[0]```
```dynamic_eps = base_eps +```
```↪ torch.abs(max_val) *```
```↪ max_val_multiplier```
```mean = x.mean(dim=1, keepdim=True)```
```var = ((x - mean)**2).mean(dim=1,```
```↪ keepdim=True)```
```rstd = torch.rsqrt(var + dynamic_eps)```
```normalized_x = (x - mean) * rstd```
```x_int = x.to(torch.int32)```
```w_selected = torch.where((x_int & 1)```
```↪ == 0, w1, w2)```
```scaled_x = normalized_x * w_selected +```
```↪ b```
```col_indices = torch.arange(n_cols,```
```↪ device=x.device)```
```is_even_col = (col_indices % 2) == 0```
```res = torch.where(```
```    is_even_col,```
```    torch.log(torch.abs(scaled_x) +```
```    ↪ log_addend),```
```    -scaled_x```
```)``` |
| Quadratic RMS Norm | A variant of RMS Norm where the standard sum of squares is replaced by a sum of the fourth power of the elements. | ```sum4 =```
```↪ x.to(torch.float32).pow(4).sum(-1,```
```↪ keepdim=True)```
```norm4 = (sum4 + eps).pow(0.25)```
```res = x * (1 / norm4).to(x.dtype) *```
```↪ weight``` |
| Triangular Norm | A custom normalization method that incorporates trigonometric functions to introduce non-linear, periodic transformations. | ```n_cols = X.shape[1]```
```col_offsets = torch.arange(0, n_cols,```
```↪ device=X.device, dtype=X.dtype)```
```perturbed_x = X +```
```↪ torch.sin(col_offsets) *```
```↪ perturbation_factor```
```chaos_mean = torch.mean(perturbed_x,```
```↪ dim=1, keepdim=True)```
```chaos_var = torch.mean(torch.cos(X) *```
```↪ torch.sin(chaos_mean), dim=1,```
```↪ keepdim=True)```
```instability_factor =```
```↪ torch.rsqrt(torch.abs(chaos_var) +```
```↪ eps)```
```normalized_row = torch.tanh((X -```
```↪ chaos_mean) * instability_factor)```
```res = torch.pow(W, 2) * normalized_row```
```↪ + torch.cos(B)``` |

| | | |
|---|---|---|
| Wormhole Norm | A custom normalization technique where the calculation for each row is perturbed by a value sampled from a different row, creating cross-row dependencies. | ```
n_rows, n_cols = x.shape
y = torch.zeros_like(x)
for i in range(n_rows):
    neighbor_row_idx = (i + 1) %
    ↪ n_rows
    neighbor_col_idx = i % n_cols
    wormhole_val = x[neighbor_row_idx,
    ↪ neighbor_col_idx].item()
    x_row = x[i, :]
    mean = x_row.mean()
    perturbed_mean = mean +
    ↪ wormhole_val *
    ↪ perturbation_factor
    var = ((x_row -
    ↪ perturbed_mean)**2).mean()
    rstd = torch.rsqrt(var + eps)
    normalized_row = (x_row -
    ↪ perturbed_mean) * rstd * w + b
    if wormhole_val > 0.5:
        y_row = normalized_row *
        ↪ wormhole_val
    else:
        y_row = normalized_row /
        ↪ (wormhole_val +
        ↪ divisor_guard)
    y[i, :] = y_row
``` |

## C.1 TLX DSL EXTENSION CASE STUDY

The following example shows a generated TLX kernel for a GEMM operator with the AlphaEvolve agent. It is the full version of the simplified example in § 4.5.

```python
import triton
import triton.language as tl
import triton.language.extra.tlx as tlx
import torch

def triton_tlx_gemm(a: torch.Tensor, b: torch.Tensor) -> torch.Tensor:
    # Check constraints
    assert a.shape[1] == b.shape[0], "Incompatible matrix dimensions"
    assert a.is_contiguous(), "Matrix A must be contiguous"
    assert b.is_contiguous(), "Matrix B must be contiguous"

    M, K = a.shape
    K_b, N = b.shape

    # Create output tensor
    c = torch.zeros((M, N), dtype=a.dtype, device=a.device)

    # Define block sizes
    BLOCK_M = 128
    BLOCK_N = 128
    BLOCK_K = 32
    NUM_STAGES = 2

    # Launch kernel
    grid = (triton.cdiv(M, BLOCK_M) * triton.cdiv(N, BLOCK_N),)

    _tlx_gemm_kernel[grid](
        a, b, c,
        M, N, K,
        a.stride(0), a.stride(1),
        b.stride(0), b.stride(1),
        c.stride(0), c.stride(1),
        BLOCK_M=BLOCK_M,
        BLOCK_N=BLOCK_N,
        BLOCK_K=BLOCK_K,
        NUM_STAGES=NUM_STAGES,
    )

    return c
```

```
864
865    @triton.jit
866    def _tlx_gemm_kernel(
867        a_ptr, b_ptr, c_ptr,
868        M, N, K,
869        stride_am, stride_ak,
870        stride_bk, stride_bn,
871        stride_cm, stride_cn,
872        BLOCK_M: tl.constexpr,
873        BLOCK_N: tl.constexpr,
874        BLOCK_K: tl.constexpr,
875        NUM_STAGES: tl.constexpr,
876    ):
877        # Program ID
878        pid = tl.program_id(axis=0)
879        num_pid_m = tl.cdiv(M, BLOCK_M)
880        num_pid_n = tl.cdiv(N, BLOCK_N)
881        pid_m = pid // num_pid_n
882        pid_n = pid % num_pid_n
883
884        # Block offsets
885        offs_am = (pid_m * BLOCK_M + tl.arange(0, BLOCK_M)) % M
886        offs_bn = (pid_n * BLOCK_N + tl.arange(0, BLOCK_N)) % N
887        offs_k = tl.arange(0, BLOCK_K)
888
889        # Initialize pointers
890        a_ptrs = a_ptr + (offs_am[:, None] * stride_am + offs_k[None, :] *
        ↪    stride_ak)
891        b_ptrs = b_ptr + (offs_k[:, None] * stride_bk + offs_bn[None, :] *
        ↪    stride_bn)
892
893        # Allocate shared memory buffers with correct dtype
894        buffers_a = tlx.local_alloc((BLOCK_M, BLOCK_K), tl.float16,
        ↪    NUM_STAGES)
895        buffers_b = tlx.local_alloc((BLOCK_K, BLOCK_N), tl.float16,
        ↪    NUM_STAGES)
896
897        # Initialize accumulator
898        acc = tl.zeros((BLOCK_M, BLOCK_N), dtype=tl.float32)
899
900        # Prefetch first stages
901        for i in range(min(NUM_STAGES - 1, tl.cdiv(K, BLOCK_K))):
902            buf_idx = i % NUM_STAGES
903            buffer_a = tlx.local_view(buffers_a, buf_idx)
904            buffer_b = tlx.local_view(buffers_b, buf_idx)
905
906            k_remaining = K - i * BLOCK_K
907            mask_k = offs_k < k_remaining
908
909            token_a = tlx.async_load(a_ptrs, buffer_a, mask=mask_k[None, :])
910            token_b = tlx.async_load(b_ptrs, buffer_b, mask=mask_k[:, None])
911            tlx.async_load_commit_group([token_a, token_b])
912
913            a_ptrs += BLOCK_K * stride_ak
914            b_ptrs += BLOCK_K * stride_bk
915
916        # Main computation loop
917        for k in range(0, tl.cdiv(K, BLOCK_K)):
           buf_idx = k % NUM_STAGES

           # Wait for current buffer to be ready
           tlx.async_load_wait_group(NUM_STAGES - 2)

           # Get buffer views
           buffer_a = tlx.local_view(buffers_a, buf_idx)
```

```
        buffer_b = tlx.local_view(buffers_b, buf_idx)

        # Perform async dot product
        acc = tlx.async_dot(buffer_a, buffer_b, acc)
        acc = tlx.async_dot_wait(NUM_STAGES - 1, acc)

        # Prefetch next buffer if available
        next_k = k + NUM_STAGES - 1
        if next_k < tl.cdiv(K, BLOCK_K):
            next_buf_idx = next_k % NUM_STAGES
            next_buffer_a = tlx.local_view(buffers_a, next_buf_idx)
            next_buffer_b = tlx.local_view(buffers_b, next_buf_idx)

            k_remaining = K - next_k * BLOCK_K
            mask_k = offs_k < k_remaining

            token_a = tlx.async_load(a_ptrs, next_buffer_a,
            ↪  mask=mask_k[None, :])
            token_b = tlx.async_load(b_ptrs, next_buffer_b,
            ↪  mask=mask_k[:, None])
            tlx.async_load_commit_group([token_a, token_b])

            a_ptrs += BLOCK_K * stride_ak
            b_ptrs += BLOCK_K * stride_bk

    # Wait for final computation to complete
    acc = tlx.async_dot_wait(0, acc)

    # Store result
    offs_cm = pid_m * BLOCK_M + tl.arange(0, BLOCK_M)
    offs_cn = pid_n * BLOCK_N + tl.arange(0, BLOCK_N)
    c_ptrs = c_ptr + stride_cm * offs_cm[:, None] + stride_cn *
    ↪  offs_cn[None, :]
    c_mask = (offs_cm[:, None] < M) & (offs_cn[None, :] < N)

    c = acc.to(tl.float16)
    tl.store(c_ptrs, c, mask=c_mask)
```

## D  LLM USAGE

We used large language models to (i) aid and polish writing (copy editing, phrasing, and shorten-
ing) and (ii) support retrieval and discovery of related work (suggesting search terms and surfacing
candidate papers). All prose, claims, equations, tables, and figures in the final manuscript were au-
thored and verified by the authors. LLM suggestions were treated as drafts and manually reviewed
and edited. For literature discovery, we independently verified every citation against the original
sources and did not rely on LLM-generated citations. LLMs were not used to design experiments,
generate results, write benchmark code, or analyze data beyond light wording suggestions.

