# OpenReview forum: "TritonGym: A Benchmark for Agentic LLM Workflows in Triton GPU Code Generation"
_ICLR.cc/2026/Conference — Submitted to ICLR 2026_

### Official Review · Reviewer_p2Ej · 2025-10-30

**Soundness:** 3
**Presentation:** 3
**Contribution:** 2
**Rating:** 4
**Confidence:** 3

**Summary:**

This article introduces TritonGym, a benchmark for evaluating llm agentic workflows on GPU kernel generation with the Triton language. TritonGym enables fair comparisons of different workflows by standardizing tools. The data includes 4 parts: maintained operators, community samples, OOD tasks and DSL extensions. Experimental results show that stronger llms can generate more reasonable kernels for familiar operators in one-shot setting but struggle with OOD and DSL tasks, while agentic workflows substantially improve both pass@1 and perf@1.

**Strengths:**

- **Clear Formulation and Motivation**: The paper identifies a real gap in existing benchmarks: current GPU code generation evaluations bundle tool access within each system, making fair comparison impossible. The motivation for standardizing tool invocation via function-call APIs is compelling and well-articulated.
- **Well-Designed Tool Space**: The set of function-call tools (compiler, verifier, evaluator, profiler, searcher, browser) is thoughtfully designed and mirrors real-world GPU kernel development workflows. The tools effectively enforce a correctness-before-performance progression and provide actionable feedback for iterative refinement.
- **OOD Operator Design**: The out-of-distribution operators are a genuinely novel contribution. Benchmark-only operators with no public implementations (e.g., Max Reduce GEMM, Chaos Norm) effectively test reasoning and optimization understanding rather than memorization, addressing a critical weakness in existing benchmarks where web search could trivialize tasks.
- **Comprehensive Model Coverage**: The evaluation includes diverse state-of-the-art LLMs from multiple vendors (Claude Sonnet 3.5, GPT-4o, DeepSeek-chat, Llama-4-Maverick) with consistent experimental settings, providing valuable insights into current model capabilities and limitations.

**Weaknesses:**

- **Missing Evaluation**: The paper positions itself as a benchmark for "agentic workflows" but only evaluates fixed, manually-designed workflows (AlphaEvolve, Geak, One-shot). It does not evaluate recent work on automated workflow generation and optimization (e.g., ADAS, AFlow, etc.).
- **Missing Tool Usage Analysis**: The paper provides no analysis of which tools are most valuable, how frequently they're invoked by different agents, or what happens when individual tools are removed. There are no ablation studies showing the impact of the profiler, searcher, or other tools on final performance. This makes it impossible to understand which components of the tool suite actually contribute to improvements.
- **Presentation Issues and Technical Errors**: The paper contains several typographical and citation errors that affect quality: Figure 3(c) has an extra closing parenthesis in the caption; Section 3.3.1 contains a broken citation marked as "(?)".

**Questions:**

- **Dataset Size**: I seem to have not seen any statement regarding the dataset size. How many operator tasks are included in the benchmark in total? How many input shape configurations are tested per operator?
- **Cost and Efficiency Metrics**: What is the computational cost (API tokens, wall-clock time, dollar cost) of different workflows? How does the cost scale with trial budget K? Can you provide cost-normalized metrics (e.g., Perf@K per dollar)?

---

> ### Author Response · Authors · 2025-11-21
>
> We thank the reviewer for recognizing the clear motivation, the design of the tool space, the OOD operator design, and the comprehensive model coverage, and for pointing out missing evaluations and presentation issues.
>
> # Missing evaluation
>
> > “The paper positions itself as a benchmark for ‘agentic workflows’ but only evaluates fixed, manually-designed workflows…”
>
> To demonstrate TritonGym as a shared testbed for agentic workflows, we include dynamic agents such as GeakAgent with **conditional tool usage**. We agree that fully automatic workflow synthesis (e.g., as in ADAS or AFlow) would be a compelling additional baseline. However, to our knowledge there is no public implementation of such systems tailored to GPU code generation. One of our goals in releasing TritonGym is precisely to enable and standardize future research on automatic workflow design in this domain.
>
> # Missing Tool Usage Analysis
>
> > “The paper provides no analysis of which tools are most valuable, how frequently they're invoked by different agents, or what happens when individual tools are removed. There are no ablation studies showing the impact of the profiler, searcher, or other tools on final performance. This makes it impossible to understand which components of the tool suite actually contribute to improvements.”
>
> We agree that understanding tool usage is crucial for studying agentic workflows, and this is exactly the kind of analysis TritonGym is designed to support. In the revised version, we will add **tool-usage and ablation statistics**, including:
>
> - For each workflow, the average number of calls per operator for each tool.
> - The fraction of tasks on which each tool is invoked.
> - Ablations that disable selected tools (e.g., profiler, searcher) and measure the impact on Perf@K.
>
> These analyses will provide a clearer picture of which tools drive improvements and how different workflows exploit the tool suite.
>
> # Presentation issues
>
> > “Figure 3(c) has an extra closing parenthesis… Section 3.3.1 contains a broken citation ‘(?)’.”
>
> We will fix the caption of Fig. 3(c), remove the broken “(?)” citation in §3.3.1, and conduct another proofreading pass to catch similar issues.
>
> # Dataset size and input-shape configurations
>
> > “How many operator tasks are included in the benchmark in total? How many input shape configurations are tested per operator?”
>
> We will reiterate in the main text that TritonGym contains **122 operator tasks** (82 standard, 13 OOD, 27 DSL-extension) and clarify the shape protocol. Each operator is associated with a set of **canonical evaluation shapes** (as in Fig. 2(a)), with at least two representative shapes per task. The probabilistic verifier materializes input tensors according to these configurations. We will add a short paragraph and a table in §3.1.1 summarizing typical numbers of shapes per operator and per category.
>
> # Cost and efficiency metrics (tokens, time, Perf@K per cost)
>
> > “Cost and Efficiency Metrics: What is the computational cost (API tokens, wall-clock time, dollar cost) of different workflows? How does the cost scale with trial budget K? Can you provide cost-normalized metrics (e.g., Perf@K per dollar)?”
>
> We appreciate this suggestion. We show the **cost statistics** such as API tokens and cost for the evaluated models. We will report these metrics in the appendix of the revised version and make them available through the leaderboard. Given the standardized harness, users can then convert these raw statistics into cost-normalized metrics (e.g., Perf@K per dollar) under their own cost models.
>
> |                   | Input (Tokens)   | Output (Tokens)  | Cost ($)   | Pass@1 | Perf@1 | Cost/Pass@1 ($) | Cost/Perf@1 ($) |
> |-------------------|---------|---------|--------|--------|--------|-------------|-------------|
> | Claude-Sonnet-4.5  | 3537422 | 8397720 | 135.13 | 36.72% | 0.326  | 368.001089  | 414.509202  |
> | GPT-5             | 3063240 | 8294007 | 86.06  | 25.39% | 0.206  | 338.952343  | 417.76699   |
> | Qwen3   | 3196531 | 8977316 | 24.08  | 40.09% | 0.404  | 60.0648541  | 59.6039604  |

---

> > ### Comment · Reviewer_p2Ej · 2025-11-24
> >
> > Thanks for the author's response.
> > - I do not see the claimed revisions reflected in the current PDF, and it is unclear whether the updates were omitted or planned for a future version.
> > - The added cost-efficiency experiments are appreciated and make the empirical evaluation somewhat more complete.
> > - Since the authors already provide implementations of various tools, adding automated-search baselines such as ADAS or AFlow should primarily require implementing the corresponding search algorithms. These baselines are still necessary for a benchmark that aims to be comprehensive.
> >
> > I therefore maintain my original score.

---

### Official Review · Reviewer_CdRk · 2025-10-30

**Soundness:** 3
**Presentation:** 3
**Contribution:** 3
**Rating:** 6
**Confidence:** 2

**Summary:**

This paper proposes TritonGym—a tool-centric benchmark for agentic workflows in Triton GPU kernel generation. Core contributions include: a standardized function-call interface (compiler/verifier/evaluator/searcher/browser/profiler) that unifies tool usage and budgets; a diverse task suite covering the maintained operator set, community samples, out-of-distribution (OOD) samples, and DSL extensions (TLX, Gluon); evaluation metrics Pass@K and Perf@K; and full logging of tool invocations to ensure reproducibility and apples-to-apples comparison. Empirically, with the same trial budget, agentic workflows outperform one-shot; gains are larger under OOD and DSL extension settings; LLM interchange orchestration can lift Pass@1 but may destabilize performance.

**Strengths:**

1. **Standardized tool interface and reproducible evaluation**: A clear function-call API with unified thresholds for compiler/verifier/evaluator, together with exhaustive logging of tool calls and outcomes, substantially improves apples-to-apples reproducibility across workflows.
2. **Comprehensive dataset and task design**: Beyond standard and community samples, TritonGym includes OOD tasks with explicit semantics but no public implementations, and DSL extension tasks for TLX and Gluon, covering a difficulty spectrum from familiarity to transfer and extension.
3. **Fairness-oriented evaluation protocol**: Pass@K and Perf@K are well defined; performance focuses on average latency while allowing custom metrics, aligning with engineering practice. Trial-budget sensitivity is also reported.
4. **Representative empirical findings**: Systematic comparisons of one-shot vs. agents, failure breakdowns, trial–performance curves, and LLM interchange orchestration substantiate the value of tool–feedback–iteration.
5. **Engineering and openness**: Docker/evaluation suite and a future leaderboard are planned; a clear roadmap targets more backends, more DSLs, and automated OOD generation.

**Weaknesses:**

**1. Positioning vs. KernelBench (Moderate)**
- Needs sharper head-to-head evidence to justify “Triton-only + standardized tool layer” versus KernelBench’s function-call and multi-language support.

**2. OOD design and anti-leakage (Major)**
- Lacks rigorous anti-leakage controls and quantification (time cutoffs, allow/deny lists, caching, visibility windows) and standard→OOD transfer analyses to evidence true generalization.

**3. Fairness and budget accounting (Moderate)**
- Unclear unified billing for compile/verify/evaluate/retrieve/profile and normalization of latency across hardware; finer-grained, leaderboard-visible accounting is needed.

**Questions:**

1. **Relation to KernelBench**: Can you provide head-to-head comparisons on overlapping tasks under the same hardware, the same trial budget, and the same tool thresholds to quantify TritonGym’s advantages (e.g., log completeness, reproducibility variance, leaderboard stability)? Why not contribute the standardized tool layer to KernelBench?
2. **OOD validity and anti-leakage**:
   - Do searcher/browser tools operate under strict time cutoffs and allow/deny lists? Is there an automated audit that enforces semantic isolation and absence of public implementations for OOD tasks?
   - Please provide standard→OOD transfer experiments and failure-mode differences to show you are measuring generalization rather than mere difficulty.
3. **Tool budget and cost accounting**:
   - Are compile/verify/evaluate/retrieve/profile calls billed into a unified budget? If so, how does the leaderboard account for and display detailed usage (a token-like ledger)?
   - Does latency measurement include warmup, repeated runs, and confidence intervals? How are hardware/driver differences normalized?

---

> ### Author Response · Authors · 2025-11-21
>
> We thank the reviewer for the positive assessment of our tool interface, dataset design, fairness-oriented evaluation, and empirical findings, and for the concrete suggestions on positioning, OOD controls, and budget accounting.
>
> # Positioning vs KernelBench and why not contribute tools there
>
> > “Can you provide head-to-head comparisons on overlapping tasks to quantify TritonGym’s advantages? Why not contribute the standardized tool layer to KernelBench?”
>
> Compared to benchmarks like KernelBench, TritonGym is designed around the **agentic workflow paradigm**. This enables us to evaluate not only final kernel performance but also the *process* by which agents interact with tools (compile, profile, retrieve, browse, refine). On overlapping tasks, we observe that agentic workflows leveraging these tools significantly outperform one-shot baselines, as shown in our experiments.
>
> We chose to build TritonGym as a new platform instead of contributing a tool layer directly to KernelBench because KernelBench is primarily a **static dataset** with evaluation scripts, whereas TritonGym is a **gym-style platform** whose core functionality is launching orchestrated workflows with multiple tool interactions. A large portion of our codebase is devoted to this orchestration layer, which does not naturally fit into KernelBench’s architecture. Instead, we import a subset of KernelBench samples to increase TritonGym’s diversity while providing the new agentic evaluation capabilities.
>
> # OOD validity and anti-leakage mechanisms
>
> > “Do searcher/browser tools operate under strict time cutoffs and allow/deny lists? Is there an automated audit that enforces semantic isolation and absence of public implementations for OOD tasks?”
>
> OOD operators in TritonGym are crafted manually by composing existing tensor primitives into novel semantics (e.g., cross-row dependencies, nonstandard normalization paths). We deliberately design these operators so that there are **no existing public implementations** of the exact semantics we benchmark.
>
> As a result, even though the searcher/browser tools can retrieve relevant background material, they cannot surface an exact implementation for these tasks. Agents therefore must **reason about the semantics**, rather than copy code. We will clarify this design principle in the paper and discuss how we audit OOD operators to avoid trivial overlaps with public code.
>
> > “Please provide standard→OOD transfer experiments and failure-mode differences to show you are measuring generalization rather than mere difficulty.”
>
> Because OOD tasks are derived from standard ones by recombining existing primitives or modifying semantics (e.g., swapping sum for max/min/product), they are not necessarily *intrinsically* harder. However, Table 1 shows that performance on OOD tasks is consistently lower than on standard tasks across all workflows, despite comparable difficulty. This gap suggests that the OOD tasks indeed measure **generalization beyond distributional similarity** rather than simply being harder instances. We will make this interpretation explicit in the revision.
>
> # Tool budget and cost accounting
>
> > “Are compile/verify/evaluate/retrieve/profile calls billed into a unified budget? If so, how does the leaderboard account for and display detailed usage (a token-like ledger)?”
>
> Currently, we **do not** unify non-LLM tool calls into a single budget metric. The cost of these tools varies dramatically with the environment: the profiler depends on GPU hardware and drivers, the browser depends on network conditions and provider, etc. Instead, our evaluation normalizes on the number of LLM calls.
>
> For users who care about non-LLM tool costs, the TritonGym harness exposes detailed logs of tool invocations, so they can compute environment-specific cost metrics (e.g., wall-clock time, GPU hours, network usage) and build their own ledgers or leaderboards. We will clarify this design choice and point to the relevant logging hooks in the documentation.
>
> > “Does latency measurement include warmup, repeated runs, and confidence intervals? How are hardware/driver differences normalized?”
>
> We measure latency by averaging **50 timed runs** after a **warmup of 10 runs**. For all reported experiments in the paper, we use a **single hardware and driver configuration** to ensure fairness. The framework itself, however, allows users to plug in different hardware/driver stacks; in that case, they can re-run the benchmark and report their own latencies. We will clarify these measurement details in the experimental setup section.

---

> > ### Comment · Reviewer_CdRk · 2025-11-24
> >
> > I thank the authors for their detailed response.
> >
> > I have also carefully read the comments from the other reviewers, which have provided valuable additional context. Given these discussions and my original assessment, I will maintain my current rating and confidence score.

---

### Official Review · Reviewer_tyNC · 2025-11-01

**Soundness:** 3
**Presentation:** 2
**Contribution:** 2
**Rating:** 4
**Confidence:** 3

**Summary:**

This work proposes a new benchmark for coding LLMs and agentic coding systems, aiming to evaluate their effectiveness at writing functional, high-performance GPU kernels in Triton (and extensions). The benchmark gathers samples from a number of sources; including a set of 'maintained' operators from open-source ML projects, other code-generation benchmarks, newly generated operators by mutating existing ones, as well as collecting samples which use language extensions. The majority of the samples appear to be taken from public repos and previously curated code-generation benchmarks. The authors additionally undertake to define an orchestration framework with the goal of better supporting the evaluation of agent systems as opposed to single-generation LLMs, and making comparisons between models fairer. This involves defining a standard set of tool signatures along with their implementation. The authors evaluate a number of models and agent systems with their benchmark, demonstrating that agentic workflows out perform one-shot solution generation and perform better on new OOD samples.

**Strengths:**

The work does a good job of collecting a range of benchmark problems. It also correctly notes that the community is faced with the need to 'migrate' a number of benchmarks, collected before agent workflows became prominent, and which make the assumption of single-shot solution generation which need to be adapted slightly for use with agent systems. The effort made to make TritonGym work with agent system evaluations is welcome. The results relating to the value of agent systems, in particular on the ability of models to tackle more challenging OOD problems are useful.

**Weaknesses:**

One slight weakness of the paper is the originality. As surveyed in the related work section, there already exist a number of GPU code generation benchmarks, many of which, especially MultiKernelBench seem to hold significant overlap with TritonGym. Looking at the data collection report in Section 3.1.2, 58% of the samples come from 'community samples' which includes public repositories and prior code-generation benchmarks. The total number of samples in the dataset is not reported either, which makes it hard to gauge the scale of the original contribution of this work.

While the effort to accommodate agent systems as opposed to single-response LLM generations is welcome, this must be done with care. It is common practice now for agent evaluations to involve the researcher or practitioner adapting early LLM benchmarks to work with their agent, although this can indeed lead to variations in prompts, submission formats and so forth. The effort to standardize the tool set is good, however as I understand it all evaluated methods must run within the same orchestration framework. My concern is that this is too restrictive for the evaluation of agent systems, since different LLMs and agents often have their preferred function calling or tool calling mechanisms, which may differ on semantics, invocation format and so forth. It would therefore be welcome if the benchmark provided a standardized definition for the tools it provides (for instance as a JSON schema) thus allowing agents to use their own tool calling mechanisms, while still leveraging the common tool implementation on the backend. I believe this would allow many more agents to be run on this benchmark with minimal modifications; just updating the available tool set. Please do clarify whether your orchestration framework makes the tool documentation (Figure 2, b) available to agents to use within their tool calling mechanisms at runtime.

**Questions:**

- what steps have you taken to ensure that all the samples within the benchmark are correct? The move from SWE-Bench to SWE-Bench Verified through rigorous (often manual) review of the questions revealed a number of poor-quality problems in the dataset. How can you guarantee that similar errors don't exist in TritonGym?
- Would it be possible to update the evaluation results table in the paper with more recent models? All the models included are previous-generation models, and there are no models included from after the end of last year which limits the usefulness of the evaluation. For instance, budget permitting, it would be good to run the evaluation on GPT-5, Sonnet 4.5, Gemini 2.5 pro, Grok 4 and perhaps also open models such as Qwen 3, DeepSeek v3, Codestral and so forth. Doing so would improve the relevance of the information the reader can derive from TritonGym on the performance of frontier LLMs, generalization to OOD tasks and the usefulness of various agent systems.

---

> ### Author Response · Authors · 2025-11-21
>
> We appreciate the reviewer’s recognition of our data collection effort and the value of adapting benchmarks to agentic systems, as well as the concerns around originality, flexibility, and correctness.
>
> # Originality and scale of contribution
>
> > “As surveyed in the related work section, there already exist a number of GPU code generation benchmarks, many of which, especially MultiKernelBench seem to hold significant overlap with TritonGym. Looking at the data collection report in Section 3.1.2, 58% of the samples come from 'community samples' which includes public repositories and prior code-generation benchmarks.”
>
> The core contribution of TritonGym is to provide a **benchmark platform for orchestrated agentic workflows** in GPU code generation. Beyond a static dataset, TritonGym offers:
>
> - A standardized **tool interface** (compiler, profiler, searcher, browser, etc.) tailored to GPU kernel development.
> - A curated set of **standard, OOD, and DSL-extension tasks** reflecting real workloads and generalization challenges.
>
> Prior benchmarks typically treat each evaluated system as a black box: the benchmark supplies input/output pairs and measures final performance. In contrast, TritonGym explicitly evaluates **the interaction between models and tools**, making it possible to study how agentic structure and tool usage affect outcomes.
>
> For the dataset itself, we carefully design for **quality**, **diversity**, and **OOD coverage**:
>
> - **Quality.** We maintain a core set of standard operators drawn from production-level codebases. These are widely used, representative workloads that appear across existing benchmarks.
> - **Diversity.** To broaden coverage, we traverse prior benchmarks and public repositories to collect community samples that are *not* covered by the maintained set.
> - **OOD coverage.** We construct novel OOD operators to stress reasoning beyond memorization given the retrieval tools available in TritonGym, by composing common primitives into new semantics (e.g., cross-row dependencies, unconventional normalization).
>
> We will clarify this positioning and the relationship to prior datasets (including MultiKernelBench) in the related-work section.
>
> # Orchestration restrictiveness and tool-calling mechanisms
>
> > “My concern is that this is too restrictive… It would be welcome if the benchmark provided a standardized definition for the tools… allowing agents to use their own tool calling mechanisms… Please clarify whether your orchestration framework makes the tool documentation available at runtime.”
>
> TritonGym’s orchestration layer is intended as a **reference harness**, not a required framework. Internally:
>
> - Each tool is defined by a JSON schema (name, signature, inputs, returns).
> - This schema is serialized into the prompt so that models see the **tool documentation at runtime**.
> - Users are free to implement their own bridges between this schema and their preferred tool-calling interface.
>
> In our codebase, we provide thin wrappers for common orchestrators (e.g., OpenAI-style and Anthropic-style tool calling). As long as the user respects the tool semantics, they can plug in a custom orchestration stack without modifying the benchmark. We will expand §3.2 to make this flexibility explicit.
>
> # Dataset correctness
>
> > “What steps have you taken to ensure that all the samples within the benchmark are correct?”
>
> For **semantic correctness**, we rely on probabilistic verification against the PyTorch oracle. GPU kernels in TritonGym have relatively shallow control flow, which makes such verification both efficient and effective. Each candidate kernel is checked on multiple randomly sampled inputs (shapes and dtypes), and we only admit samples that pass verification with high confidence.
>
> For **dataset quality**, we:
>
> - Manually maintain a standard operator set whose Triton kernels come from production-level codebases and are regularly tested.
> - Traverse existing benchmarks and repositories to collect additional community samples that are not covered by the maintained set.
>
> We will expand Section 3.1 to describe this pipeline more systematically.

---

> > ### Author Response · Authors · 2025-11-21
> >
> > # Frontier models and recency
> >
> > > “Would it be possible to update the evaluation results table in the paper with more recent models? ”
> >
> > Yes. We evaluate several recent models with results shown as following. The results demonstrates a similar trend as the observations discussed in Section 4. Beyond the camera-ready, we will continue to maintain a **public leaderboard** that tracks new models over time.
> >
> > |                   | Standard Pass@1 | Standard Perf@1 | OOD Pass@1 | OOD Perf@1 | DSL Pass@1 | DSL Perf@1 | Full Pass@1 | Full Perf@1 |
> > |-------------------|-----------------|-----------------|------------|------------|------------|------------|-------------|-------------|
> > | One-shot          |                 |                 |            |            |            |            |             |             |
> > | GPT-5             | 26.10%          | 0.260           | 15.40%     | 0.104      | 8.30%      | 0.026      | 16.60%      | 0.130       |
> > | Claude-Sonnet-4.5 | 55.10%          | 0.610           | 23.10%     | 0.223      | 16.70%     | 0.089      | 31.63%      | 0.307       |
> > | Qwen3             | 50.00%          | 0.466           | 23.10%     | 0.305      | 25.00%     | 0.077      | 32.70%      | 0.282       |
> > | Geak              |                 |                 |            |            |            |            |             |             |
> > | GPT-5             | 29.60%          | 0.285           | 23.10%     | 0.132      | 25.00%     | 0.109      | 25.90%      | 0.175       |
> > | Claude-Sonnet-4.5 | 53.50%          | 0.577           | 23.10%     | 0.269      | 25.00%     | 0.119      | 33.87%      | 0.321       |
> > | Qwen3             | 56.40%          | 0.597           | 23.10%     | 0.317      | 25.00%     | 0.095      | 34.83%      | 0.336       |
> > | AlphaEvolve       |                 |                 |            |            |            |            |             |             |
> > | GPT-5             | 45.20%          | 0.443           | 30.80%     | 0.404      | 25.00%     | 0.087      | 33.67%      | 0.311       |
> > | Claude-Sonnet-4.5 | 69.50%          | 0.754           | 61.50%     | 0.872      | 33.30%     | 0.124      | 54.77%      | 0.583       |
> > | Qwen3             | 63.80%          | 0.501           | 30.80%     | 0.479      | 33.30%     | 0.094      | 42.63%      | 0.358       |

---

### Official Review · Reviewer_di7i · 2025-11-01

**Soundness:** 3
**Presentation:** 3
**Contribution:** 3
**Rating:** 4
**Confidence:** 3

**Summary:**

The paper introduces TritonGym, a benchmark and orchestration framework for evaluating agentic LLM workflows in Triton GPU kernel generation. The key contribution is standardizing tool access via a function-call API, decoupling model capability from workflow design and enabling fair apples-to-apples evaluation. The benchmark includes curated operators, community samples, OOD operators, and DSL extensions (e.g., TLX, Gluon), with extensible hardware descriptors and backend support. Empirical results show (i) one-shot LLMs perform reasonably on common operators but fail on OOD and DSL tasks, and (ii) agentic workflows significantly improve correctness and performance.

**Strengths:**

1. **OOD operator design**: The creation of benchmark-only operators (e.g., max-reduce GEMM, chaos norm) that test reasoning rather than memorization is creative and addresses the retrieval leakage problem intelligently.

2. **Extensibility framework**: The explicit backend and DSL descriptor architecture enables systematic evaluation across diverse hardware and language extensions.

3. **Comprehensive baseline coverage**: Evaluation of 4 frontier LLMs across 3 workflow paradigms (one-shot, Geak, AlphaEvolve) with detailed failure analysis provides thorough empirical grounding.

4. **Practical relevance**: Addresses real adoption challenges for LLM-generated GPU kernels with extensible backend/DSL support.

**Weaknesses:**

1. **Insufficient scale & statistical rigor**:  The paper does not explicitly report the total number of benchmark samples. No confidence intervals or significance tests despite probabilistic metrics.

2. **Weak OOD validation**: Claims OOD tasks "require reasoning beyond memorization" lack empirical support.  No systematic analysis of OOD similarity to training data. There is no analysis of OOD operators' similarity to training distributions, nor evaluation against retrieval-augmented baselines. As a result, the degree to which OOD tasks genuinely measure generalization remains unclear.


3. **Insufficient dataset quality assurance and documentation**：The dataset construction and oracle validation process lacks transparency. Quality control criteria for DSL extensions and specialized operators is not clear. OOD operators correctness and quality control ('handcraft efficient Triton implementations as baselines' in line 218) are unclear. No discussion of how oracle implementations were validated or tested.   This raises concerns about oracle reliability and the validity of performance comparisons.

4. The manuscript contains an unresolved editing placeholder (“shallow control flow (?)” Line 297)

**Questions:**

1. OOD validation: How do you measure OOD similarity to training data? Can you empirically show that retrieval-augmented generation fails on OOD tasks, demonstrating non-triviality?

2. Hardware specs usage: Do models actually exploit hardware specifications (Figure 2d) in generated code? Can you show examples?

3. Trial budget: In Figure 5, how do you ensure fair comparison between K independent generations vs. K iterative refinements (same total LLM calls)?


4. Oracle quality: How were oracle implementations validated? Could Perf@1 > 1.0 indicate oracle weaknesses?

---

> ### Author Response · Authors · 2025-11-21
>
> We thank the reviewer for highlighting the strengths of TritonGym (OOD design, extensibility, baseline coverage, and practical relevance) and for the detailed suggestions on scale, rigor, and documentation. Below we address each concern and summarize the concrete changes we will make.
>
> # Insufficient scale and statistical rigor
>
> > “The paper does not explicitly report the total number of benchmark samples. No confidence intervals or significance tests despite probabilistic metrics.”
>
> **Scale and composition.** We will clarify the size and breakdown of TritonGym and make these statistics more prominent in the main text. The benchmark currently contains **122 operator tasks**: **82 standard** (35 maintained, 47 community), **13 OOD**, and **27 DSL-extension** tasks (12 TLX, 15 Gluon). This breakdown is shown in Figure 3(a). TritonGym is a living benchmark; these numbers reflect the state at submission time. We emphasize that we maintain a high-quality core of production-inspired standard operators and augment them with community operators that broaden coverage beyond this core.
>
> **Statistical analysis.** We do not run full significance tests across all baselines because of the substantial computational cost: each evaluation already involves multiple shapes, dtypes, and probabilistic verification runs. However, we will improve the discussion of statistical stability. In particular, Figure 5 shows the probabilistic sampling trend of Perf@K for different agentic workflows, using a fixed sampling temperature of 0.1. We will clarify that we repeated each configuration with multiple random seeds, report the variance in the appendix, and highlight that all key qualitative conclusions are robust under this variability.
>
> # Weak OOD validation and similarity to training data
>
> > “Claims OOD tasks ‘require reasoning beyond memorization’ lack empirical support… No analysis of OOD similarity to training distributions or evaluation against retrieval-augmented baselines.”
>
> Existing GPU code-generation benchmarks focus almost exclusively on operators with public implementations, which makes it difficult to disentangle memorization from reasoning—especially for heavily optimized kernels such as GEMMs and FlashAttention. In TritonGym, agentic workflows are further equipped with **retrieval tools** (searcher and browser) that can access external resources, which amplifies this challenge.
>
> To address this, TritonGym introduces OOD operators (e.g., Max-Reduce GEMM, Chaos Norm, Wormhole Norm) that are **designed from first principles** by altering standard operators in ways that are unlikely to appear in public repositories.
> Section C describes the underlying transformations (e.g., replacing sums with max/min/product, introducing cross-row coupling). We will make this design protocol more explicit in the main text and add a short discussion of why these constructions are unlikely to appear in public repositories, thereby strengthening the case that performance on OOD tasks reflects generalization rather than memorization.
>
> # Dataset QA and documentation (OOD, DSL, oracles)
>
> > “The dataset construction and oracle validation process lacks transparency… No discussion of how oracle implementations were validated or tested.”
>
> TritonGym oracles come from two sources:
>
> 1. **Maintained operators.** For the core standard set, we use hand-written implementations drawn from production-level codebases and maintained by our team.
> 2. **Community operators.** For additional diversity, we collect representative implementations from public repositories and prior benchmarks.
>
> For *correctness*, we validate all oracles against the PyTorch reference using probabilistic verification over multiple shapes and dtypes. For *quality*, we will expand Section 3.1 to describe the provenance of maintained vs. community operators, our review process for community kernels, and the tests we run before admitting an operator into the benchmark.
>
> # Editing placeholder
>
> > “The manuscript contains an unresolved editing placeholder (‘shallow control flow (?)’).”
>
> We apologize for this leftover placeholder. In the final version we will replace it with a proper citation to typestate verification, for example:
>
> `John Field et al., 2003. *Typestate verification: abstraction techniques and complexity results.* International Conference on Static Analysis.`

---

> > ### Author Response · Authors · 2025-11-21
> >
> > # Hardware specs usage
> >
> > > “Do models actually exploit hardware specifications (Figure 2d) in generated code? Can you show examples?”
> >
> > Yes. Figure 7 shows a pipelined GEMM operator that uses the Tensor Core `mma` instruction on NVIDIA Hopper GPUs. Without hardware specifications, the model would not know about this instruction and could not emit correct `mma`-based code. Designing an *efficient* pipeline, however, also requires overlapping `mma` with asynchronous memory operations (e.g., `async_load`), which in turn depends on latency information from the profiler. This example illustrates why **combining hardware specifications with profiling tools** is important, and we will explicitly reference this example when discussing Figure 2(d).
> >
> > # Trial budget
> >
> > > “In Figure 5, how do you ensure fair comparison between K independent generations vs. K iterative refinements (same total LLM calls)?”
> >
> > In Figure 5, we align the compared workflows by the **number of LLM calls used for code generation**. This guarantees that “K independent generations” and “K iterative refinements” use the same total LLM generation budget. We agree that other cost metrics—such as total computation, token usage, or API dollar cost—are also meaningful. Our goal here is to isolate the *agentic structure* of the workflows under a fixed, fine-grained budget. We will clarify this in the text and note that users can easily extend the provided logging tools to normalize by additional cost metrics.
> >
> > # Oracle weakness
> >
> > > “How were oracle implementations validated? Could Perf@1 > 1.0 indicate oracle weaknesses?”
> >
> > For correctness, we validate each oracle using the probabilistic verifier across multiple shapes and dtypes. For performance, we use production-level implementations for highly optimized operators (e.g., GEMMs, FlashAttention) in the maintained set, and representative community implementations elsewhere. Thus, the oracle performance represents the **best-known practical implementation** in our corpus, but not necessarily the absolute optimum.
> >
> > Consequently, cases where Perf@1 > 1.0 indicate that the generated kernel exceeds our oracle on the given hardware and shape. This can happen if (i) a strong agent finds an optimization that surpasses human baselines, or (ii) the oracle is suboptimal on a particular corner-case input or hardware. In either case, the oracle is intended as a **reference baseline**, and Perf@K is interpreted relative to that baseline rather than as a proof of global optimality.

---

### Meta-Review · Area_Chair_qLat · 2026-01-05

**Summary:**

Across reviewers, the main concerns center on three issues: (1) novelty and positioning, given overlap with prior GPU kernel benchmarks and the fact that much of the dataset is collected from existing sources; (2) rigor and validity of the benchmark, especially OOD design, anti-leakage guarantees, oracle quality, and statistical reporting; (3) completeness of evaluation for an “agentic workflow” benchmark, including missing automated workflow baselines, limited tool-usage analysis, and cost accounting. While reviewers agree the motivation and engineering are solid and practically relevant, there is disagreement on whether these contributions are sufficient for acceptance at ICLR.

**Reviewer Concerns:**

1. Reviewer di7i highlighted strengths in OOD operator design, extensibility, and baseline coverage, but raised substantive concerns on statistical rigor, OOD validation, oracle reliability, and dataset documentation. The rebuttal addressed most points with concrete clarifications on dataset size, validation procedures, budget alignment, and examples of hardware-spec usage, which largely resolve the technical concerns.
2. Reviewer tyNC focused on originality and flexibility. Concerns about overlap with existing benchmarks and restrictiveness of orchestration were partly mitigated by clarifications that TritonGym provides tool schemas and allows custom orchestration. However, the originality concern remains only partially addressed, as the dataset itself still heavily overlaps prior work.
3. Reviewer CdRk was more positive overall, emphasizing reproducibility and fairness, but raised major concerns on OOD anti-leakage and budget accounting. The rebuttal clarified design intent but did not introduce concrete quantitative anti-leakage audits or unified cost ledgers, leaving some concerns outstanding.
4. Reviewer p2Ej emphasized missing evaluations (automated workflow generation), lack of tool ablations, and presentation errors. While the authors promised additional analyses and cost statistics, the reviewer noted these were not visible in the current revision and maintained the score.

Overall, rebuttals were detailed and thoughtful, and several technical clarifications strengthen the work. However, concerns around novelty relative to existing benchmarks and completeness as a benchmark for agentic workflows are not fully resolved.

**Reviewer Scores:**

1. Reviewer di7i: Likely unchanged. Core technical concerns were addressed, but the reviewer’s original stance was marginally below threshold.
2. Reviewer tyNC: Likely unchanged. Clarifications improve flexibility but do not fundamentally change the originality assessment.
3. Reviewer CdRk: Explicitly stated they will maintain the original rating.
4. Reviewer p2Ej: Explicitly stated they will maintain the original rating.

---

### Decision · Program_Chairs · 2026-01-26

Reject